# A Latent Profile Analysis of PERMA: Associations with Physical Activity and Psychological Distress among Chinese Nursing College Students

**DOI:** 10.3390/ijerph192316098

**Published:** 2022-12-01

**Authors:** Meiling Qi, Qian Sun, Xiangyu Zhao, Yiming Gao, Di Zhao, Shiyu Shen, Polat Zayidan, Ziyang Xiu, Ping Li

**Affiliations:** 1School of Nursing and Rehabilitation, Cheeloo College of Medicine, Shandong University, Jinan 250012, China; 2School of Nursing, College of Xinjiang Uyghur Medicine, Hetian 848099, China; 3School of Physical Education, Shandong University, Jinan 250012, China; 4Jinan Vocational College of Nursing, Jinan 250102, China

**Keywords:** PERMA, exercise, mental health, college students, latent profile analysis

## Abstract

Background: The wellbeing of college students is an important concern for public health, and may have associations with insufficient physical activity and psychological distress. This study aimed to identify the latent classes of wellbeing based on the PERMA (i.e., positive emotions, engagement, relationships, meaning, and accomplishments) wellbeing framework, and to explore their associations with levels of physical activity and psychological distress. Methods: A cross-sectional online survey was conducted. A latent profile analysis was performed to characterize the different classes of wellbeing of nursing college students. Results: A group of 1741 nursing college students in China completed the study. Three wellbeing classes were identified in the final model (i.e., low-level wellbeing, moderate-level wellbeing, and high-level wellbeing). Significant differences were found between the three classes in terms of gender (*p* = 0.002) and year of study (*p* = 0.038). Low levels of physical activity participation were significantly associated with lower odds of being in the high-level wellbeing class compared with the moderate-level wellbeing class (OR = 1.398, 95%CI [1.023, 1.910], *p* = 0.035). Lower levels of psychological distress were also associated with greater wellbeing among the three wellbeing classes (*p* < 0.05). Conclusions: Effective strategies are needed to increase college students’ physical activity participation and decrease the severity of psychological distress to improve their health and wellbeing in China.

## 1. Introduction

Nursing college students often experience high learning pressure, and such life patterns can influence their wellbeing (e.g., psychological distress and depression) [1,2]. Other potential factors, such as a lack of correct family education, increasing fierce competition after graduation, and serious exposure to internet use also adversely affect college students’ mental health [3,4]. In this respect, studies have reported that more than 30% of college students suffer from psychological distress [5]. Furthermore, the existing literature has found that negative wellbeing and mental health are associated with reduced academic function, frustration, poor physical health conditions, and even a high risk of suicidal tendencies [5,6]. One previous study has also described that nursing students experience a higher prevalence of wellbeing disorders than students in other subjects [7]. Therefore, it is important to understand the multidimensional wellbeing of nursing college students in order to develop well-targeted public health interventions and policies and improve their wellbeing.

A growing body of empirical evidence suggests that wellbeing is mainly characterized by positive emotions, engagement, relationships, meaning, and accomplishments constituting a theoretical framework referred to as PERMA [8,9]. Specifically, positive emotions refer to the affective component or feeling well, with engagement referring to a deep psychological connection to a particular activity, relationship referring to social connection, meaning referring to purpose, and accomplishments referring to personal goals [10,11]. Wellbeing and flourishing have been examined using the PERMA wellbeing framework in adult workers [12], cancer survivors [13], and artists [14] across countries such as Korea, China, Brazil, and India. The PERMA wellbeing framework was also used in university contexts to examine the associations between wellbeing and the online learning environment [15], as well as health education [16]. However, these studies considered a single-dimensional approach in explaining the scores of five elements of PERMA wellbeing. Because the PERMA wellbeing profiler is used to calculate the mean scores of the five PERMA wellbeing elements, but not the categorical scores to classify different levels of wellbeing, limited studies have used the PERMA to explore different classes of wellbeing in nursing college students. To address this limitation, latent profile analysis (LPA) was suggested to be used to examine how the five PERMA wellbeing elements interact and to classify different classes of wellbeing of the nursing college students.

Previous studies have suggested that decreasing the psychological distress of nursing college students is critical for prevention and intervention to enhance their overall wellbeing [17]. This is because psychological distress (e.g., anxiety and depression) may contribute to an increased rate of mental disorders that have implications for nursing college students’ educational and career development [1]. In addition, protective factors, such as physical activity participation, are considered as effective approaches to improve nursing college students’ wellbeing [18]. Individuals who engage in regular physical activity are more likely to have an increased sense of wellbeing and a reduction in psychological distress [19]. Given the importance of developing differential interventions to enhance college students’ wellbeing, it is necessary to examine the associations between physical activity participation and psychological distress and different wellbeing classes in this population. Therefore, the first aim of this study was to use the LPA to identify classes of wellbeing based on the five-element PERMA wellbeing framework in Chinese college students. In addition, this study examined the impact of levels of physical activity participation and psychological distress on different classes of wellbeing in this population. This study hypothesized that there would be significant differences in physical activity participation and levels of psychological distress among different wellbeing classes.

## 2. Materials and Methods

### 2.1. Study Design

A cross-sectional study using an online survey design was performed in this study. Ethics approval for this study was received from the University Human Research Ethics Committee (2021-R-165). Completion and submission of the online survey implied consent to participate. This was declared to respondents at the commencement of the survey.

### 2.2. Participants and Recruitment

Using convenience sampling, nursing students from a large medical college in China were invited to participate. All nursing students who were willing to participate in this study were eligible to enroll. An email invitation with the help of the College Deputy Vice Administration Office was sent to all nursing students. The email invitation included the purpose of the study, inclusion and exclusion criteria, and the online survey link (https://www.wjx.cn (accessed on 10 October 2021)) to the questionnaire. A total of 1741 Chinese nursing college students were included for analysis in this study after removing those who reported more than 960 mins of total physical activity or sedentary time per day (n = 184, 9.6%). Participants were aged between 17 and 24 years (M = 19.38; SD = 1.02). Among these participants, most were female (n = 1366, 78.5%), and were in their second year of study (n = 765, 43.9%).

### 2.3. Data Collection

Data collection took place between 10 October and 30 December 2021. The study survey included participants’ demographic details (i.e., age, gender, and year of study), PERMA-Profiler, International Physical Activity Questionnaire—Short Form (IPAQ-SF), and an assessment of psychological distress.

The PERMA-Profiler Chinese version was used to evaluate individuals’ multidimensional wellbeing [20]. This study used the 15 items of the PERMA-Profiler to assess the five elements of wellbeing (i.e., positive emotions, engagement, relationships, meaning, and accomplishments). Three items assessed each PERMA element, and composite scores were averaged across the three items per element. Each item was scored on a Likert-type scale ranging from 0 to 10 (0 = not at all, 10 = completely; 0 = never, 10 = always; 0 = terrible, 10 = excellent), with higher scores indicating greater wellbeing. This tool demonstrated good internal consistency among Chinese nursing college students in this study (Cronbach’s α = 0.933).

The IPAQ-SF Chinese version was used to assess participants’ physical activity participation and average sitting time on weekdays and weekends during the past seven days [21]. There are two types of IPAQ scores for data processing and analysis: a categorical and a continuous score. The categorical score classified participants into three physical activity intensity levels (i.e., low, moderate, and high). The continuous score is expressed as the metabolic equivalent task (MET minutes per week) of energy expenditure. In addition, participants’ sitting time (i.e., hours per day) was also recorded on the IPAQ-SF. High validity and reliability for the IPAQ-SF have been established among Chinese adults with intraclass correlation coefficients above 0.84 [21].

The 10-item Kessler Psychological Distress Scale (K10), Chinese version, was used to assess psychological distress in the past month [22]. The K10 is a self-reported questionnaire containing ten questions with a score ranging from 1 to 5 to assess participants’ frequency of nonspecific psychological distress across the past month based on questions related to symptoms of anxiety and depression. Participants chose how often they felt or thought in a certain way: 1 = almost never, 2 = sometimes, 3 = fairly often, 4 = very often, and 5 = all the time. The total score was obtained by summing all 10 items, with a total score of 10–50. A score of 15 or less reflected no symptoms of distress, while low distress ranged from 16 to 21, moderate distress ranged from 22 to 29, and high distress ranged from 30 to 50. The K10 scale is a valid instrument with acceptable internal consistency, with Cronbach’s α over 0.954 in this study.

### 2.4. Statistical Analysis

Data analysis was conducted using the Statistical Package for the Social Sciences (SPSS) version 27.0 and the Mplus 8.3. Based on the data-cleaning rules for the IPAQ-SF, respondents who reported over 960 min of total physical activity or sedentary time per day were identified as over-reporting. The assumption is that individuals spend an average of 8 h of sleep per day [23]. Descriptive statistics were calculated using frequencies (i.e., percentages) for categorical variables and mean and standard deviations for continuous variables.

The LPA was used to determine the optimal number of wellbeing classes based on the PERMA wellbeing framework among Chinese nursing college students. Starting from the initial model (one category), the number of categories in the model is gradually increased until the model fits the data optimally. The optimal model was determined by a comprehensive consideration of fit indicators and theoretical values. The model fit indicators are the Log-likelihood test, Akaike’s information criterion (AIC), Bayesian information criterion (BIC), sample-size-adjusted Bayesian information criterion (ssaBIC), entropy (>0.8 is acceptable), and Lo–Mendell–Rubin (LMR), and the bootstrapped likelihood ratio test (BLRT). The smaller AIC, BIC, and ssaBIC are more desirable and represent the models with better fit and which are more parsimonious. Additionally, the sample size requirement was considered in the LPA, with a sample size of more than 5% in each class.

Differences in demographic characteristics and the five PERMA elements among the wellbeing classes within the final model were compared using analysis of variance (ANOVA) (i.e., age, PERMA elements) and chi-square tests (i.e., gender and year of study). Furthermore, multinomial logistics regression analyses were conducted to assess the association of levels of physical activity participation and psychological distress with the latent classes. Covariates (i.e., gender, year of study) with *p* values of <0.05 were included in the multinomial logistics regression. The significance level was set at 0.05.

## 3. Results

A majority of the participants had engaged in either moderate (n = 881, 50.6%) or high (n = 385, 22.1%) levels of physical activity in the past week. Almost half of the participants had reported no psychological distress in the past month (n = 769, 44.2%). The average scores of participants’ psychological distress scores and sitting time per day during workdays and weekdays were 18.67 (SD = 8.0), 6.56 (SD = 2.80), and 5.41(SD = 3.08), respectively. Additionally, participants’ average scores of the five PERMA wellbeing elements were more than 6.26 (SD range = 1.89–2.11) for each element (see Table 1).

Table 2 displays the fit statistics of latent classes of the PERMA wellbeing of Chinese nursing college students. LL, AIC, BIC, and adjusted BIC continued to decrease with an increase in the number of latent classes. The LMR test was not significant for the 5-class, indicating that the 5-class solution did not fit the data better than the 4-class solution. The model identification and entropy value indicated that the 3- or 4-class models were most suitable. This study selected a 3-class model considering the sample size of each class being > 5%, a high entropy value, and higher model identification in a 3-class model than a 4-class model. Based on the parsimony and interpretability of the classes, a 3-class model was selected.

There are significant differences in the five PERMA wellbeing elements across the three latent classes (*p* < 0.001) (see Table 3). The 1-class (5.4%) was labeled as low-level wellbeing, with the lowest mean scores of the five PERMA elements (M range = 1.91–2.42, SD range = 1.39–1.81). The 2-class made up 51.2% of the overall sample and was labeled as moderate-level wellbeing, as the mean scores of the five elements of PERMA in class 2 were in a middle level (M range = 5.31–6.22, SD range = 1.03–1.28). The 3-class (43.4%) was labeled as high-level wellbeing, with the highest mean scores of the five PERMA elements (M range 7.69–8.70, SD range = 1.02–1.36). The latent classes of PERMA wellbeing are shown in Figure 1.

Table 3 also describes differences in the three wellbeing classes on participants’ characteristics. No significant differences were found between the three classes in terms of age. However, compared to participants with low-level wellbeing, participants with moderate- and high-level wellbeing were more likely to be females (78.13 and 80.53 vs. 64.89%, *p* = 0.002), and less likely to be in the second year of study (43.18 and 44.47 vs. 46.81%) and third year of study (23.34 and 27.76 vs. 29.79%).

Table 4 shows the associations between levels of physical activity participation and psychological distress with the three wellbeing classes after adjusting for covariates (i.e., gender and year of study). When comparing the high-level wellbeing class relative to the moderate-level wellbeing class, having no distress (OR = 0.109, 95%CI [0.075, 0.160], *p* < 0.001) or low distress (OR = 0.296, 95%CI [0.199, 0.440], *p* < 0.001) was significantly associated with higher odds of being in the high-level wellbeing class. Adversely, engaging in a low level of physical activity was significantly associated with lower odds of being in the high-level wellbeing class (OR = 1.398, 95%CI [1.023, 1.910], *p* = 0.035).

When comparing the low-level wellbeing class relative to the high-level wellbeing class, physical activity participation did not differ significantly between the two classes. However, there was a trend that no distress (OR = 0.131, 95%CI [0.071, 0.242], *p* < 0.001), low distress (OR = 0.208, 95%CI [0.103, 0.419], *p* < 0.001) and moderate distress (OR = 0.408, 95%CI [0.188, 0.886], *p* = 0.024) were associated with increased odds of being in the high-level wellbeing class relative to the low-level wellbeing class.

## 4. Discussion

To our knowledge, this is the first study that has classified the latent classes of wellbeing using the PERMA wellbeing framework and measured associations between these wellbeing classes and levels of physical activity participation and psychological distress in Chinese nursing college students. Three latent classes were identified among nursing college students, including the low-level wellbeing class, moderate-level wellbeing class, and high-level wellbeing class.

This study demonstrated a significant difference in gender among the three wellbeing classes, which is in line with recent studies that the level of wellbeing among male students was lower than that among female students [24,25]. One potential reason suggested was that females report a higher frequency of online health information searching for health data and nutrition information, while males are more interested in smoking and online games via online health applications. Importantly, a higher percentage of participants in their second and third year of study seem to experience low-level wellbeing compared to moderate- and high-level wellbeing. This finding is not consistent with a similar study which indicated that senior students may accumulate more health information and enhanced health awareness than junior students to promote their health and wellbeing [24]. Specific reasons for these differences may be attributed to different major backgrounds of students and assessment tools of wellbeing. The implementation of clinical curriculum programs for nursing students in their second and third year of study may also contribute to the high prevalence of low-level wellbeing because of the risk of failure for nursing students in a professional clinical course [26]. In addition, increasing fierce competition after graduation may be a potential factor reducing the second-year and third-year nursing college students’ wellbeing [3]. Considered together, these findings suggest the implementation of health education among nursing college students may be an effective strategy to improve their wellbeing.

Although studies have demonstrated the benefits of physical activity on the health and wellbeing of college students [19,27], there was an increase in the physical inactivity and sedentary time in this demographic found in the current study. This study found that 27.3% of college students reported lower levels of physical activity participation. There are several potential factors that may contribute to insufficient physical activity in college students, such as prolonged sitting time, low awareness of the beneficial effects of physical activity, and lack of time and motivation [19,27]. This study also found that the average sitting time of college students was more than 6 h per day during workdays. Similar prolonged sedentary sitting time was also found in individuals in the university workplaces [28,29]. Previous studies have indicated that insufficient physical activity participation has the potential to decrease academic engagement and behavior among college students, and prolonged sitting time is also associated with high levels of psychological distress [30,31]. It is, therefore, important to increase levels of physical activity participation and decrease sedentary sitting time in the design of effective interventions for the health and wellbeing of nursing college students.

This study found that low levels of physical activity participation were associated with low levels of wellbeing among the moderate- and high-level wellbeing classes of nursing college students. The positive association between low levels of physical activity participation and wellbeing implies that insufficient physical activity participation may be a potential risk factor in developing wellbeing. Low physical activity participation may contribute to a low level of quality of life, academic performance, and mental health, which are positively associated with wellbeing [31,32]. Furthermore, a sedentary lifestyle was positively associated with students’ inactivity and long screen time, which have been identified as major public health issues that increase the risk of mental health problems (e.g., depression and anxiety) of college students [33]. Knowledge of the association between physical activity participation and wellbeing could help target interventions and direct resources to the college students [19]. Given the potential adverse effects of insufficient physical activity, there is a need to pay more attention to those nursing college students who engage in low levels of physical activity.

This study also found that moderate-to-high psychological distress accounted for 31.4%, which is to some extent in accordance with results of previous studies which found almost 24–35% of college students suffered from psychological issues, such as depression and anxiety [10,24]. A high level of psychological distress can result in a decline in college students’ quality of life, which may contribute to low levels of wellbeing [19]. However, only 5.1% of the nursing students reported low-level wellbeing in this current study. These findings suggest that other beneficial factors may reduce the risk of psychological distress and then positively influence nursing students’ wellbeing; for example, a high percentage of participants had moderate and high levels of physical activity participation in this current study (72.7%). Interestingly, compared with physical activity participation, the severity of psychological distress may be a more serious factor that influences college students’ wellbeing, as significantly negative associations between levels of psychological distress and the three latent classes were found. The results of the relationship between psychological distress and wellbeing are consistent with recent studies which indicated that negative emotion had a direct influence on early adults’ PERMA wellbeing elements [34,35]. This suggests that stress disorder syndrome was a potential predictor of low sleep health and sleep behavior, which may in turn influence wellbeing and quality of life. Another study also indicated that psychological distress can have an influence on the psychosocial functioning of quality of life and further reduce wellbeing [19]. Hence, relevant programs to reduce college students’ psychological distress are recommended to enhance their health and wellbeing in future studies.

Some limitations of this study should be considered. Firstly, some basic information about students, such as family background and education, were not investigated in this study, as students’ family background may have the potential to influence their wellbeing as indicated in the introduction chapter. Secondly, the gender distribution, with most respondents being females in this study, also limits the generalizability of the study results. Thirdly, using self-reported measures may also be a study limitation, as self-reported outcomes can lower the accuracy of the data and further affect the latent class results. Fourth, this current study is a convenience sample of nursing college students and is not representative of college students. Fifthly, this study enrolled nursing students, and group differences in the study variables between nursing students and students in other subjects are still unclear. Future studies might need to be conducted to compare different groups of wellbeing classes of students in other subjects and nursing and examine their associations with psychological distress and physical activity.

## 5. Conclusions

The LPA identified three classes of wellbeing based on the five elements of the PERMA wellbeing framework among Chinese nursing college students, including low-level wellbeing, moderate-level wellbeing, and high-level wellbeing. Considering the gender differences as the covariation, low levels of physical activity participation were associated with low levels of wellbeing between the moderate-level and high-level wellbeing classes. Furthermore, there were negative correlations between the levels of psychological distress and wellbeing among the three wellbeing classes of Chinese nursing students. These findings suggest that practice strategies to increase nursing college students’ physical activity participation and reduce the severity of their psychological distress may be effective to promote their health and wellbeing.

## Figures and Tables

**Figure 1 ijerph-19-16098-f001:**
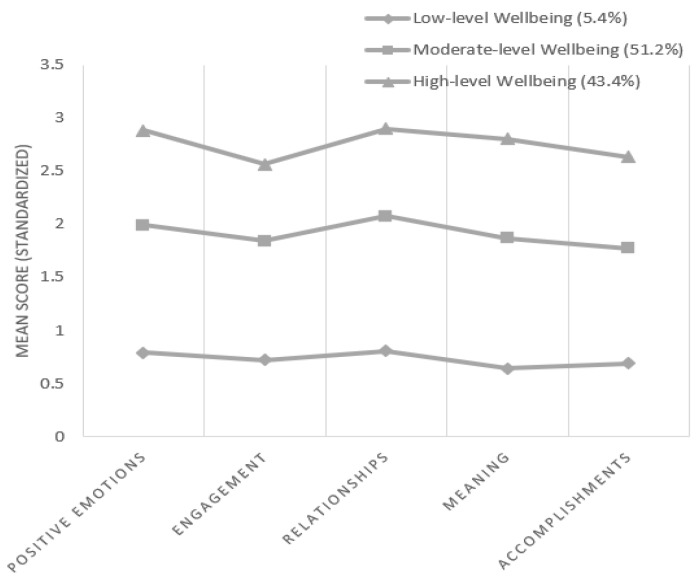
Standardized mean symptom scores by latent classes of wellbeing.

**Table 1 ijerph-19-16098-t001:** Demographic characteristics of participants (n = 1741).

Variables	M ± SD/N (%)
**Age**	19.38 ± 1.02
**Gender**	
Male	375 (21.5%)
Female	1366 (78.5%)
**Year of study**	
First year	530 (30.4%)
Second year	765 (43.9%)
Third year	446 (25.6%)
**Physical activity participation**	
LPA	475 (27.3%)
MPA	881 (50.6%)
HPA	385 (22.1%)
**Average Sitting time per day on workdays(h/day)**	6.56 ± 2.80
**Average sitting time per day on weekdays(h/day)**	5.41 ± 3.08
**Psychological distress**	18.67 ± 8.00
No distress	769 (44.2%)
Low distress	425 (24.4%)
Moderate distress	301 (17.3%)
High distress	246 (14.1%)
**Positive emotion**	6.92 ± 2.05
**Engagement**	6.28 ± 1.89
**Relationship**	7.10 ± 2.05
**Meaning**	6.63 ± 2.11
**Accomplishments**	6.26 ± 2.00

Notes: M = mean; SD = standard deviation; PA = physical activity; LPS = low physical activity; MPA = moderate physical activity; HPA = high physical activity.

**Table 2 ijerph-19-16098-t002:** Fit statistics of latent classes of PERMA (n = 1741).

Model	K	LL	AIC	BIC	ssaBIC	Entropy	LMR	BLRT
C1	10	−8903.553	17,827.105	17,881.727	17,849.958			
C2	16	−6847.837	13,727.675	13,815.070	13,764.240	0.859	<0.0001	<0.0001
**C3**	**22**	**−5644.624**	**11,333.248**	**11,453.416**	**11,383.525**	**0.923**	**0.0193**	**<** **0.0001**
C4	28	−4807.077	9670.153	9823.095	9734.142	0.909	0.0004	<0.0001
C5	34	−4491.196	9050.392	9236.107	9128.092	0.885	0.2359	<0.0001
C6	40	−4221.735	8523.469	8741.958	8614.881	0.892	0.0033	<0.0001

Notes: C = class; LL = log likelihood; AIC = Akaike’s information criterion; BIC = Bayesian information criterion; ssaBIC = Sample size adjusted Bayesian information criterion; LMR = Lo–Mendell–Rubin; BLRT = bootstrapped likelihood ratio test; bold letters indicate the best-fitting models.

**Table 3 ijerph-19-16098-t003:** Comparison of demographic characteristics and PERMA elements by latent classes (n = 1741).

Variables	Low-Level Wellbeing	Moderate-Level Wellbeing	High-Level Wellbeing	*F*/χ2	*p*
**Age**	19.49 ± 1.11	19.34 ± 1.04	19.41 ± 0.98	1.708	0.182 ^a^
**Gender**					
Male	33 (35.11%)	194 (21.87%)	148 (19.47%)	12.215	**0.002 ^b^**
Female	61 (64.89%)	693 (78.13%)	612 (80.53%)		
**Year of Study**					
First-year	22 (23.40%)	297 (33.48%)	211 (27.76%)	8.269	**0.038 ^b^**
Second-year	44 (46.81%)	383 (43.18%)	338 (44.47%)		
Third-year	28 (29.79%)	207 (23.34%)	211 (27.76%)		
**Positive emotion**	2.18 ± 1.63	5.59 ± 1.10	8.65 ± 1.02	2154.291	**<** **0.001 ^a^**
**Engagement**	2.16 ± 1.66	5.51 ± 1.03	7.69 ± 1.36	1211.396	**<** **0.001 ^a^**
**Relationship**	2.42 ± 1.81	6.22 ± 1.28	8.70 ± 1.08	1544.464	**<** **0.001 ^a^**
**Meaning**	1.91 ± 1.39	5.60 ± 1.14	8.40 ± 1.15	2018.922	**<** **0.001 ^a^**
**Accomplishments**	2.04 ± 1.61	5.31 ± 1.06	7.90 ± 1.26	1604.456	**<** **0.001 ^a^**

Notes: ^a^ Value was calculated by one-way ANOVA; ^b^ value was calculated by a chi-square test; Bold numbers indicate significant differences (*p* < 0.05).

**Table 4 ijerph-19-16098-t004:** Multinomial logistic regression analysis of levels of PA participation and psychological distress on latent classes (n = 1741).

Variables		High-Level vs. Moderate-Level Wellbeing			High-Level vs. Low-Level Wellbeing		
		*p*	OR	95%CI	*p*	OR	95%CI
**PA participation (Ref. = HPA)**							
	LPA	**0.035**	1.398	(1.023, 1.910)	0.247	1.341	(0.787, 2.539)
	MPA	0.467	1.107	(0.841,1.458)	0.366	0.818	(0.447, 1.346)
**Psychological distress (Ref. = High distress)**							
	No distress	**<** **0.001**	0.109	(0.075, 0.160)	**<** **0.001**	0.131	(0.071, 0.242)
	Low distress	**<** **0.001**	0.296	(0.199, 0.440)	**<** **0.001**	0.208	(0.103, 0.419)
	Moderate distress	0.244	0.768	(0.492, 1.198)	**0.024**	0.408	(0.188, 0.886)

Notes: PA = physical activity; LPS = low physical activity; MPA = moderate physical activity; HPA = high physical activity; Ref. = reference; analyses were adjusted for significant covariates (i.e., gender); Bold numbers indicate significant differences (*p* < 0.05).

## Data Availability

The data presented in this study are available on request from the corresponding author.

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
