# Peer review of "A Latent Profile Analysis of PERMA: Associations with Physical Activity and Psychological Distress among Chinese Nursing College Students"

_ijerph, 2022, doi:10.3390/ijerph192316098_

Round 1

Reviewer 1 Report

It should be noted that the authors of this manuscript have done a lot of work. However, there are a number of points that can be improved. The relationship of subjective well-being with regular physical activity and psychological distress has been studied for a long time. However, these studies (since Argyle's monograph) are not reflected in the literary review.

In the paragraph "Participants" it is necessary to reflect all the data about the sample.

It does not become clear from the study what the authors found new in comparison with the data that were obtained earlier in the studies of psychologists. The specifics of the sample are also unclear: how can students of a medical college, for example, differ from, for example, a construction one?

Unfortunately, the study does not reveal socio-psychological or psychological-pedagogical (at least any) reasons for a high level of psychological distress from moderate to high in 31.7% of cases, with only about 5.4% of people with a low level of subjective well-being.

Therefore, it would be desirable to try to interpret the results from the position of reflecting the psychological mechanisms of the studied connection in relation to the specifics of the sample. Perhaps a typological analysis could provide more information about the specifics of the relationship between activity, distress and well-being of students.

Reviewer 2 Report

This study aimed to the latent classes of Chinese nursing college students in China who exhibited similar well-being patterns based on PERMA framework (5 domains: Positive emotions, Engagement, Relationships, Meaning, and Accomplishments), and to examine how the subgroups differed in demographic variables (age, gender, and year of study), psychological distress and physical activity. Cross-sectional data was collected from 1,741 nursing college students in China. Latent profile analysis was used to identity 3 classes: low-level, moderate-level, and high-level wellbeing. Being female, a first-year student, less psychological distress and greater physical activity were found to be associated with high and/or moderate well-being memberships. Even though this manuscript is well written and organized, the major finding should not be surprising given the three memberships which were classified low-moderate-high well-being subgroups. It would be expected to think of naming the memberships based on 5 PERMA domains if possible such as positive emotion and low meaning……something like that. What will be of great interest is to further differentiate these 3 subgroups by the levels of physical activity, and then examine how the subgroups differed in demographic variables (age, gender, and year of study). By doing either option as the above, it would be more interesting. In addition, as psychological distress appears to overlap with one of well-being domains-- positive emotions. As such, it could be expected the relationship between LPA memberships and psychological distress. Additional comments are outlined below.

1.          Line 56, would expect to clarify why prior well-being studies in nursing college students were considered a single-dimensional approach?

2.          To use abbreviations, would expect to write out the terms on first use, and then followed by the abbreviations in parentheses such as PERMA, IPAQ.

3.          Would expect to provide the reliability and validity of IPAQ-SF in the current study. Line 112: intraclass correlation coefficients above 0.84 on IPAQ-SF looks like to the result for the 20th reference

4.          Line 228-233, Discussion: wondering if professional clinical curriculum/ course programs in 2nd and 3rd years of nursing school could influence the well-being of nursing college students?  

Round 2

Reviewer 1 Report

First of all, I would like to express my gratitude for the changes made by the authors to the text. The manuscript has become more informative. Despite the remaining questions about the specifics of the sample, I believe the article can be useful for assessing the ratio of physical activity and psychological distress, which can be used for more effective organization of the school day and the implementation of physical activity programs for future nurses.

Author Response

We have added the specific details of the sample in the “PARTICIPANTS” section, including age, gender, and their year of study as “Participants were aged between 17 and 24 years (M = 19.38; SD = 1.02). Among these participants, most were female (n = 1,366, 78.5%), and were at their second year of study (n = 765, 43.9%).” at L97-99.

Reviewer 2 Report

Thank for this opportunity to review this revised manuscript. This revision has been significantly improved. The only remaining comment I have with the abstract is that it would be better to mention the statistical approach of latent profile analysis. After this is addressed, I would recommend the manuscript for publication.

Author Response

We have added “The latent profile analysis was performed to characterize different classes of wellbeing of nursing college students” at L21-22 in the abstract.